# Scorpion-Venom-Derived Antimicrobial Peptide Css54 Exerts Potent Antimicrobial Activity by Disrupting Bacterial Membrane of Zoonotic Bacteria

**DOI:** 10.3390/antibiotics9110831

**Published:** 2020-11-20

**Authors:** Jonggwan Park, Jun Hee Oh, Hee Kyoung Kang, Moon-Chang Choi, Chang Ho Seo, Yoonkyung Park

**Affiliations:** 1Department of Bioinformatics, Kongju National University, Kongju 38065, Korea; for_quality@naver.com (J.P.); chseo@kongju.ac.kr (C.H.S.); 2Department of Biomedical Science, Chosun University, Gwangju 61452, Korea; toqkfqkekr2@naver.com (J.H.O.); mgenetics@daum.net (H.K.K.); choist777@gmail.com (M.-C.C.)

**Keywords:** antimicrobial peptide, venom, antibiotics, zoonosis disease, feed additive

## Abstract

Antibiotic resistance is an important issue affecting humans and livestock. Antimicrobial peptides are promising alternatives to antibiotics. In this study, the antimicrobial peptide Css54, isolated from the venom of *C. suffuses*, was found to exhibit antimicrobial activity against bacteria such as *Listeria monocytogenes*, *Streptococcus suis*, *Campylobacter jejuni*, and *Salmonella typhimurium* that cause zoonotic diseases. Moreover, the cytotoxicity and hemolytic activity of Css54 was lower than that of melittin isolated from bee venom. Circular dichroism assays showed that Css54 has an α-helix structure in an environment mimicking that of bacterial cell membranes. We examined the effect of Css54 on bacterial membranes using *N*-phenyl-1-naphthylamine, 3,3′-dipropylthiadicarbbocyanine iodides, SYTOX green, and propidium iodide. Our findings suggest that the Css54 peptide kills bacteria by disrupting the bacterial membrane. Moreover, Css54 exhibited antibiofilm activity against *L. monocytogenes*. Thus, Css54 may be useful as an alternative to antibiotics in humans and animal husbandry.

## 1. Introduction

The development and emergence of multidrug-resistant bacteria are a serious crisis for humans and livestock worldwide [1]. Many countries have monitored antibiotic use to decrease antibiotic resistance [2]. Because of antibiotic-resistant infections, annual global deaths are expected to increase from 700,000 in 2014 to 10 million by 2050. Most antibiotics are used in animal husbandry to protect against infection, treat ill animals, and promote growth [3,4]. Antibiotic-resistant bacteria used in animal husbandry spread to humans in various ways, such as through food, contaminated water, and soil [5]. Zoonotic diseases are those transferred from animals to infect humans, threatening the life and health of people worldwide [6]. The World Health Organization (WHO) reports the number of diseases originating from contaminated food. The Interagency Food Safety Analytics Collaboration described *Salmonella typhimurium* and *Listeria monocytogenes* as priority foodborne pathogens [7]. *Listeria monocytogenes* and *Salmonella typhimurium* that originate in animals can cause bacterial diseases such as listeriosis and salmonellosis, which are major bacterial zoonosis diseases [8].

*Listeria monocytogenes* is a Gram-positive bacterium found in various environments such as water, soil, and extreme environments, including low and high temperatures, high salt concentrations, and a broad pH range. *Listeria monocytogenes* infects immunocompromised persons, pregnant women, and neonates, resulting in abortion, meningitis, and blood poisoning [9]. *Salmonella typhimurium* is a Gram-negative and zoonotic pathogenic bacterium. *Salmonella typhimurium* causes gastrointestinal diseases associated with morbidity and mortality rates. This bacterium negatively affects animal husbandry because it infects young animals, resulting in sepsis, vomiting, fever, and abortion. Many antibiotics are added to animal feed of animals used for food to prevent zoonotic diseases. This has increased the antibiotic resistance of bacteria and become a global issue [10]. Preventing and controlling zoonotic diseases is crucial for the development of the livestock industry and maintaining public health [11]. Thus, alternatives to antibiotics should be used to kill bacteria without inducing resistance.

Venoms produced by different species, such as scorpions, snakes, and spiders, are used as protection or to capture prey. Various peptides found in venom produced by scorpions have been reported to have broad-spectrum antimicrobial, anticancer, and anti-inflammatory activities [12,13]. Numerous scorpion venom antimicrobial peptides (AMPs) have been reported as various therapeutic agents because of their low resistance rate [14,15]. Therefore, venoms are an important source of biologically active molecules and potential therapeutic agents, including AMPs.

AMPs produced by various organisms function as effector molecules to defend against pathogen infection [16]. AMPs commonly comprise 12–50 amino acids and show antibiotic activities against bacteria, fungi, and yeast, as well as anti-inflammatory activities. AMPs possess a net positive charge property, enabling them to bind to the bacterial membrane through electrostatic and hydrophobic interactions and penetrate the bacterial membrane to induce bacterial lysis [17]. Antimicrobial mechanisms of AMPs against bacteria are largely divided into two groups. First, AMPs can disrupt the bacterial membrane. Second, AMPs can cross the bacterial membrane without causing disruption to act as intracellular components [18]. AMPs are also less prone to developing resistance than currently used antibiotics [19]. In a previous study, the novel antimicrobial peptide Css54, isolated from the venom of scorpion *Centruroides suffuses*, was studied for its antimicrobial activity against *Escherichia coli* and *Staphylococcus aureus*. Moreover, the synergistic effect of Css54 with commercial antibiotics against *S. aureus* was evaluated [20].

In this study, Css54 is tested for its antimicrobial activity against *L. monocytogenes*, *S. typhimurium*, *Streptococcus suis*, and *Campylobacter jejuni*, which cause zoonotic disease. We used melittin, an antimicrobial peptide found in bee venom. We confirmed that Css54 has an antimicrobial activity similar to melittin. We tested the cytotoxicity of Css54 against PK(15) cells and hemolytic activities using pig kidney and sheep blood, respectively. Moreover, we examined its antibiofilm activity against *L. monocytogenes* and confirmed that antibacterial activity was maintained at various salt concentrations, temperatures, and pH values. The mechanism of action of Css54 against *L. monocytogenes* and *S. typhimurium* was studied by examining outer membrane permeability, membrane depolarization, and integrity of the bacterial membrane after treatment with Css54. Our findings show that Css54 can be used as an alternative to antibiotics to treat bacterial zoonosis.

## 2. Results

### 2.1. Peptide Structure and Characterization

Css54 was isolated from *C. suffusus* venom. The amino acid sequence, molecular weight, and net charge of Css54 are listed in Table 1. Css54 consists of 25 amino acids. The net charge value and hydrophobicity were 5 and 0.532, respectively. A wheel diagram and three-dimensional structure analysis showed that the hydrophobic and hydrophilic regions formed an α-helix structure (Figure 1A,B). Css54 synthesis and molecular weights were confirmed by reversed-phase high-performance liquid chromatography (RP-HPLC) using a C18 column and matrix-assisted laser desorption ionization time-of-flight mass spectrometry (MALDI-TOF MS; Appendix A).

### 2.2. Circular Dichroism Measurements

The secondary structure of Css54 was measured by circular dichroism (CD) spectroscopy in various concentrations of sodium dodecyl sulfate (SDS) to mimic the negative charge of the bacterial membrane and trifluoroethanol (TFE) solution to mimic the hydrophobic environment of the bacterial membrane [21,22]. In 10 mM sodium phosphate, mimicking an aqueous environment, Css54 displayed a random coil (Figure 2A). In contrast, Css54 exhibited an α-helix conformation in membrane-mimicking environments, depending on the TFE and SDS concentrations (Figure 2B,C). These results indicate that Css54 can affect the bacterial membrane through electrostatic and hydrophobic interactions.

### 2.3. Antimicrobial Activities against Zoonotic Pathogens

The antimicrobial activities of peptides and antibiotics against zoonotic pathogens are summarized in Table 2. We used melittin isolated from bee venom as a control, which is known as a lytic peptide with strong antibacterial activity and used in various antibiotics for animals. Css54 showed broad-spectrum antimicrobial activity against *L. monocytogenes*, *S. typhimurium*, *Campylobacter jejuni*, and *Streptococcus suis*, with minimum inhibitory concentrations (MICs) ranging from 2 to 4 μM. These MIC values are similar to those of melittin. The highest MIC values for cefotaxime, vancomycin, and colistin were more than 64 μM. These results suggest that Css 54 effectively inhibited zoonotic bacterial growth.

### 2.4. Cytotoxicity and Hemolytic Activities of Antimicrobial Peptides

To measure the toxicity of Css54, we tested hemolytic activity by measuring hemoglobin, which appears red in the supernatant of sheep red blood cells (sRBCs) after treatment with the peptide. Melittin, which was used as a control, induced more than 80% hemolysis at 16 μM. However, the hemolytic percentage of Css54 was approximately 40% at 16 μM (Figure 3A). To investigate the cytotoxicity of AMPs against mammalian cells, we performed an MTT assay. Css54 exhibited lower cytotoxicity than melittin at a concentration of 2.5 μM; melittin treatment resulted in cell survival rates of approximately 10%. Css54 displayed a cell viability rate of approximately 50% at 8 μM. These results indicate that Css54 exhibited lower cytotoxicity than melittin (Figure 3B).

### 2.5. Antimicrobial Activities in Various Environments and Stability against Heat

To assess the antimicrobial activity of Css54 under various *L. monocytogenes* growth conditions, Css54 was tested by MIC assay at pH values of 4, 6, 8, and 10, temperatures of 4–40 °C, and salt concentrations of 0–8%. At different pH values, Css54 at 0.5× and 1× MIC inhibited the growth of *L. monocytogenes* compared to the control (Figure 4A). Moreover, Css54 showed antimicrobial activity at various temperatures (Figure 4B). In the salt sensitivity test, the antimicrobial activity of Css54 was maintained in the presence of different salt concentrations (Figure 4C). Moreover, Css54 retained its antimicrobial activity after heating at 100 °C for 80 min (Figure 4D). These results suggest that the antimicrobial activity of Css54 can be maintained under the growth conditions of *L. monocytogenes* and showed thermal stability for 80 min.

### 2.6. Biofilm Inhibition Assay

*Listeria monocytogenes* can attach to many food surfaces and form biofilms [23]. Therefore, the inhibition of biofilm formation by *L. monocytogenes* strains by Css54 was assessed. The antibiofilm activities of Css54 and melittin were compared. Melittin is known to have antibiofilm activity [24,25]. Css54 and melittin (both at 4 μM) inhibited biofilm formation by *L. monocytogenes* (KCTC 3710) by more than 50% (Figure 5A). For the strains *L. monocytogenes* (KCCM 43155) and *L. monocytogenes* (KCCM 40307), Css54 and melittin at 2 μM inhibited biofilm formation by more than 50% (Figure 5B,C).

To visualize bacteria in the biofilm, we used SYTO9, a green dye that stains live cells. After treatment with Css54 and melittin at 1, 2, and 4 μM, *L. monocytogenes* (KCTC 3710), *L. monocytogenes* (KCCM 43155), and *L. monocytogenes* were viable within the biofilm (KCCM 40307), respectively, compared to the control (Figure 5D–F). These images were consistent with the quantitative measurement of biofilms by crystal violet staining and indicate that Css54 has a strong antibiofilm activity, similar to melittin against *L. monocytogenes.*

### 2.7. Outer Membrane Permeability and Membrane Depolarization Assay

We examined the disruption of the outer membrane of *S. typhimurium* after treatment with Css54 at 0.5×, 1×, and 2× MIC in an *N*-phenyl-1-naphthylamine (NPN) uptake assay. NPN shows increased fluorescence intensity in hydrophobic environments [26]. The fluorescence intensity increased in a dose-dependent manner after treatment with Css54 (Figure 6A). These data suggest that the hydrophobic environment increased as the outer membranes were disrupted by Css54, resulting in an increase in fluorescence intensity.

The effect of Css54 on membrane potential was investigated using DiSC_3_(5). DiSC_3_(5) is distributed in the cytoplasmic membrane of bacteria, where self-quenching occurs. When the bacterial membrane is damaged by antimicrobial agents, the dye is released into the medium, resulting in increased fluorescence intensity [27]. After completely stabilizing the fluorescence intensity of DiSC_3_(5), *L. monocytogenes* and *S. typhimurium* were treated with Css54. In *S. typhimurium*, Css54 MIC showed a fluorescence intensity of approximately 30 min after 30 min (Figure 6B). In *L. monocytogenes*, Css54 at 2× MIC showed a fluorescence intensity of approximately 110 after 30 min (Figure 6C). The fluorescence intensity increased in a dose-dependent manner, indicating that Css54 induced membrane depolarization of *S. typhimurium* and *L. monocytogenes*.

### 2.8. Effect of Peptides on Membrane Integrity

The integrity of the bacterial membrane was determined using SYTOX green and propidium iodide (PI). Neither dye can pass through the intact membrane. However, if the bacterial membrane is disrupted by antimicrobial agents, these dyes bind to DNA, resulting in an increase in fluorescence intensity. Css54 at 0.5, 1, and 2× MIC at 30 min showed fluorescence intensities of approximately 31, 65, and 93 in *S. typhimurium* (Figure 7A). However, the control group of *S. typhimurium* showed a fluorescence intensity of approximately 4. Css54 at 0.5, 1, and 2× MIC increased fluorescence intensity by approximately 17, 23, and 43 at 30 min in *L. monocytogenes*, whereas this value in the control group of *L. monocytogenes* was approximately 11 (Figure 7B). In *S. typhimurium* and *L. monocytogenes*, treatment with Css54 immediately increased the fluorescence intensity in a dose-dependent manner compared to the control sample in both strains.

We further analyzed the integrity of the bacterial membranes by PI staining and flow cytometry. In *S. typhimurium*, the percentage of PI staining was approximately 16% in the control group. After treatment with Css54 at 0.5× and 1× MIC, approximately 21% and 75% of bacteria were stained with PI (Figure 7C). In *L. monocytogenes*, the percentage of PI staining was approximately 6% in the control group. Following treatment with Css54 at 0.5× and 1× MIC, around 57% and 77% of bacteria were stained with PI (Figure 7D). Css54 dose-dependently induced an increase in PI-positive bacteria. Collectively, these data show that Css54 killed bacteria by disrupting the membranes of *L. monocytogenes* and *S. typhimurium*.

## 3. Discussion

The overuse of antibiotics has rapidly increased the development of antibiotic-resistant bacteria and become a serious concern worldwide [28]. Many antibiotics are globally used on animals used as food, which is a major source of antibiotic-resistant bacteria development [29]. Thus, new antibacterial agents have been developed to kill antibiotic-resistant bacteria without causing resistance. AMPs have been considered as substitutes for antibiotics. AMPs play important roles as host defense molecules against infection in all living organisms [30]. AMPs show broad-spectrum activity against many strains of Gram-positive and Gram-negative bacteria through different modes of action, such as membrane disruption and via the intracellular target model [31,32]. AMPs have been reported to be less likely to develop drug resistance and have an advantageous effect on nutrient sources and the gut microbiota in animals [29,33]. Therefore, AMPs are good alternatives to the antibiotics used in humans and animal husbandry.

Venoms of various species are beneficial sources of bioactive molecules, such as AMPs [14]. In this study, we used melittin isolated from bee venom as a control peptide, as it is a representative lytic peptide known to have strong antimicrobial activity [34,35]. In a previous study, Css54 isolated from *C. suffusus* showed antimicrobial activity and synergistic effects with rifampicin, which is used to treat tuberculosis. Moreover, Css54 displayed an α-helical structure in 60% TFE [20]. However, how Css54 kills bacteria and whether it has antimicrobial activity against other bacteria, except for *S. aureus* and *E. coli*, remains unclear. Therefore, we focused on the antimicrobial activity and mechanism of action of Css54 against bacterial pathogens in zoonosis.

Zoonotic diseases are caused by pathogens such as bacteria, fungi, and viruses that transmit between animals and humans [36]. The WHO report in 2015 warned of the number of diseases caused by food contaminated with pathogenic bacteria such as *Salmonella* sp., *Listeria* Sp., *Campylobacter* sp., and the Enterobacteriaceae family [37]. We examined the antimicrobial activity of Css54 against bacteria on the zoonosis announced by the WHO; *S. suis* is a zoonotic pathogen that is related to swine infection [38]. Antimicrobial assays showed that Css54 exhibited antimicrobial activity against four strains similar to melittin. Moreover, the antimicrobial activity of conventional antibiotics such as cefotaxime, vancomycin, colistin, and ampicillin was evaluated. Css 54 exerted antimicrobial activity with low MIC compared to antibiotics against some strains. In a previous study, treatment with melittin showed no adverse effects on stomach tissue, including its function, and melittin was confirmed to have a protective effect against gastric inflammation and antitumor effects in vivo [39,40,41]. These reports indicate that Css54 can be developed as an effective treatment because of its lower toxicity (versus melittin). We confirmed that hemolytic activity towards sRBCs and cytotoxicity against pig kidney cells PK(15) was lower than that of melittin. The MIC value of Css54 ranged from 2 to 4 μM, and hemolytic activity and cytotoxicity following treatment with 16 μM Css54 (approximately 4-fold of the MIC value) were approximately 10% and 40%, respectively. Therefore, Css54 has the potential for treating bacterial diseases in animals and humans.

*Listeria monocytogenes* can survive in various typical environments, such as food processing and preservation [42]. This species can grow over a wide pH range of 4.5–9.6 and a temperature range of 2–45 °C. Moreover, *L. monocytogenes* can survive in a high-salt environment, with up to 10% NaCl [43]. Thus, *L. monocytogenes* is frequently present in animal foods and can contaminate foods. In addition to the biofilm inhibitory effect of Css54, one of the requirements for food and feed additives is stability under these conditions. Interestingly, Css54 maintained antimicrobial activity over wide pH, temperature, and NaCl ranges. Additionally, the processing of many foods and feeds involves a heating step, and thus antimicrobial agents must exhibit thermal stability [44]. Css54 showed antimicrobial activity after incubation for 80 min at 100 °C. These data demonstrate the potential of Css54 in applications such as food and feed additives.

Biofilm is a self-produced matrix of extracellular polymeric substances produced by microorganisms. Biofilms are important for the growth and survival of bacteria under various conditions, including low temperature, high salt concentrations, and low pH [45]. *Listeria monocytogenes* is a problem in food safety because it can form biofilms upon contact with food surfaces and persist in food processing environments [23]. Therefore, food additives used in the food industry should have antibiofilm activity. Recently, food additives have gradually become one of the most important antimicrobial agents in the food industry. Our data showed that Css54 can inhibit biofilm formation by *L. monocytogenes*, similar to melittin.

AMPs display secondary structures such as α-helices, β-sheets, loop types, and mixed structures [46]. CD spectra were determined to analyze the secondary structure of the peptides. AMPs showed a random coil structure in aqueous solution and formed a secondary structure in membrane-mimicking environments. We evaluated the secondary structure of Css54 in aqueous solution as well as bacterial membrane-mimicking environments such as SDS and TFE solutions. Css54 formed a random coil in an aqueous solution. In contrast, Css54 has an α-helix structure in SDS, which mimics the negatively charged bacterial membrane, and in TFE, which can help AMPs induce secondary structure and mimic the hydrophobic environment [47]. These results suggest that Css54 exerts its antimicrobial activity by interacting with the bacterial membrane.

To determine the antimicrobial mechanism, we examined the effect of Css54 on bacterial membranes in an outer membrane permeability assay using NPN dye and a cytoplasmic membrane depolarization assay using DiSC_3_(5); we also examined membrane integrity using SYTOX green and PI. Css54 significantly increased NPN fluorescence intensity and DiSC_3_(5) fluorescence intensity. Our results demonstrate that Css54 disrupts the outer membrane of *S. typhimurium* and induces membrane depolarization of *L. monocytogenes* and *S. typhimurium*. We examined the integrity of the bacterial membrane after treatment with Css54 using SYTOX green, PI, and SYTOX green. Fluorescence intensity increased when Css54 was added to the bacteria. This result was confirmed by flow cytometry using PI. Treatment with Css54 increased the PI fluorescence intensity, indicating that the bacterial membrane was disrupted by Css54. Our data show that Css54 has antimicrobial activity and a mechanism similar to melittin, but with lower cytotoxicity than melittin.

## 4. Materials and Methods

### 4.1. Materials

Thiazolyl blue tetrazolium bromide (MTT), *N*-phenyl-1-naphthylamine (NPN), 3,3′-dipropylthiadicarbocyanine iodide (DiSC_3_(5)), SYTOX green, propidium iodide (PI), dimethyl sulfoxide (DMSO), cefotaxime, colistin, and ampicillin were purchased from Sigma-Aldrich (St. Louis, MO, USA). Vancomycin was purchased from LPS Solution (Deajeon, Korea). Sheep red blood cells (sRBCs) were purchased from MB Cell (Seoul, Korea).

### 4.2. Microorganisms

*Salmonella typhimurium* (ATCC 14028) was purchased from the American Type Culture Collection (ATCC, Manassas, VA, USA). *Listeria monocytogenes* (KCTC 3710), *Streptococcus suis* (KCTC3557), and *Campylobacter jejuni* (KCTC 5327) were purchased from the Korea Collection for Type Cultures. *L. monocytogenes* (KCCM 40307) and *L. monocytogenes* (KCCM 43155) were purchased from the Korean Culture Center of Microorganisms (KCCM, Seoul, Korea)*. Salmonella typhimurium* (CCARM 8009) and *S. typhimurium* (CCARM 8013) were purchased from the Culture Collection of Antibiotics Resistant Microbes (CCARM, Seoul, Korea). PK(15) isolated from porcine kidney was purchased from the Korean Cell Line Bank (Seoul, Korea).

### 4.3. Peptide Synthesis and Sequence Analysis

Peptides were synthesized using the solid-phase-9-fluorenylmethocycarbonyl1 (F-moc) method on a Rink amide-4-methylbenzhydrylamine resin and a Liberty microwave peptide synthesizer (CEM Co., Matthews, NC, USA). Hydroxybenzotriazole (0.1 M) dissolved in piperidine and dimethylformamide and 0.45 M 2-(1H-benzotriazole-1-yil)-1,1,3,3-tetramethyluroniunm hexafluorophosphate dissolved in dimethylformamide were used as linkage reagents. After washing with dichloromethane, cleavage was performed in a solution of trifluoroacetic acid, phenol, water, and triisopropylsilane for 2 h at 25 °C. The crude peptides were diluted with ice-cold ether for precipitation. The peptides were added to tubes and then completely dried. The dried peptides were diluted in distilled water and then purified by RP-HPLC on a Jupiter C18 column (4.6 × 250 mm, 300 Å, 5 μm; Phenomenex, Torrance, CA, USA). The molecular weights of the peptides were confirmed by MALDI-TOF MS (Kratos Analytical, Inc., Chestnut Ridge, NY, USA). Projections of the predicted three-dimensional structures were confirmed using PEP-FOLD3 (https://bioserv.rpbs.univ-paris-diderot.fr/services/PEP-FOLD3/). PEP-FOLD3 can be used for structural characterization of the peptides, followed by visualization using PyMOL [48,49]. The HeliQuest site (http://heliquest.ipmc.cnrs.fr) was used to generate helical wheel diagrams.

### 4.4. Antimicrobial Activity Assay

To investigate the antimicrobial activity of AMPs and antibiotics, the minimum inhibitory concentrations (MICs) of peptides and antibiotics were determined using the broth dilution method [50]. *Listeria monocytogenes* (KCTC 3710), *L. monocytogenes* (KCCM 40307), *L. monocytogenes* (KCCM 43155), and *S. suis* (KCTC 3557) were cultured overnight at 37 °C in brain heart infusion (BHI). *Salmonella typhimurium* (KCTC 14028), *S. typhimurium* (CCARM 8009), *S. typhimurium* (CCARM 8013), and *C. jejuni* (KCTC 5327 were cultured overnight at 37 °C in Mueller Hinton broth (MHB). These bacteria were diluted in appropriate media to a final concentration of 2 × 10^5^ CFU/mL. AMPs and antibiotics were diluted at concentrations of 0.5.64 μM in 10 mM sodium phosphate buffer (Sp buffer, pH 7.2) in 96-well plates. Next, a 50-μL aliquot of bacteria was added to 50 μL of diluted peptide and antibiotics and incubated at 37 °C for 16–24 h. The growth of bacteria was measured as the absorbance at 600 nm using a microplate reader. MIC values were determined as the lowest concentration that inhibited bacterial growth compared to the control without AMPs and antibiotics.

### 4.5. Hemolytic Activity

The hemolytic activity of the peptides was determined using sRBCs. sRBCs were centrifuged at 2000× *g* for 10 min at 4 °C and washed three times with phosphate-buffered saline (PBS, pH 7.4). The washed sRBCs were diluted in PBS to a final concentration of 8%. AMPs (100 μL) were serially diluted from 1 to 16 μM in 96-well plates, and then sRBCs were subsequently mixed for 1 h at 37 °C. The reacted samples in the 96-well plate were centrifuged at 1500× *g* for 10 min. The supernatant was transferred to a new 96-well plate, and the absorbance was measured at 414 nm (A_414_). Next, 0.1% TritonX-100 and PBS were used as negative and positive controls, respectively. The percentage of hemolysis was calculated as follows [51]:% Hemolysis = [(A_414_ in the peptides solution − A_414_ in PBS)/(A_414_ in 0.1% TritonX-100 − A_414_ in PBS)] × 100

### 4.6. Cytotoxicity Activity

The cytotoxicity of AMPs towards PK(15)-isolated porcine kidney cells was determined by MTT assay. PK(15) cells were cultured in Dulbecco’s modified Eagle medium (DMEM) supplemented with 10% fetal bovine serum and 1% penicillin at 37 °C with 5% CO_2_. The cells were seeded into 96-well plates at 2 × 10^4^ cells/well and incubated for 24 h. AMPs were added to the wells at concentrations ranging from 0.5 to 16 μM and incubated for 24 h. Next, 20 μL of 0.5 mg/mL MTT was added to the wells and incubated for 4 h. The supernatant was removed, and 100 μL DMSO was added to dissolve the formazan crystals. The absorbance was measured at 570 nm (A570) using a microplate reader. The positive control was incubated only with DMEM. The percent cell viability was calculated according to the following formula:% Cell viability = (A_570_ in peptides solution/A_570_ in positive control) × 100

### 4.7. Stability of Peptides

The stability of Css54 was determined in different environments with varying pH values, temperatures, and salt concentrations. To confirm the effect of the antimicrobial activity of Css54 at various pH values, Sp buffer was adjusted with NaOH and HCl to pH values of 4, 6, 8, and 10. Css54 at 0.5× and 1× MIC was diluted in the prepared buffer and incubated with *L. monocytogenes* at 2 × 10^5^ CFU/mL in BHI medium for 12 h. The growth of *L. monocytogenes* was determined by measuring the absorbance at 600 nm. To evaluate the stability of Css54 at various temperatures, *L. monocytogenes* at 2 × 10^5^ CFU/mL in BHI medium was incubated with Css54 at 1× MIC at 4, 20, 30, and 40 °C for 30 min. The suspensions were plated on BHI agar plates. Colonies were counted after overnight incubation. To determine the effect of Css54 at various salt concentrations, *L. monocytogenes* at 2 × 10^5^ CFU/mL in BHI medium supplemented with 2%, 4%, 6%, and 8% NaCl was incubated with Css54 at 1× MIC overnight. The growth of *L. monocytogenes* was determined by measuring the absorbance at 600 nm. To confirm heat resistance, Css54 at a final concentration of 1× MIC was incubated for different times (20, 40, 60, and 80 min) at 100 °C and cooled on ice. Thereafter, *L. monocytogenes* at 2 × 10^5^ CFU/mL was added to the suspension and incubated overnight. The growth of *L. monocytogenes* was determined by measuring the absorbance at 600 nm.

### 4.8. Biofilm Inhibition Assay

To confirm biofilm inhibition by AMPs, *L. monocytogenes* was cultured in BHI at 37 °C. Bacterial suspensions at a final concentration of 5 × 10^5^ CFU/mL were diluted in BHI supplemented with 0.2% glucose. Bacterial suspensions (90 μL) and peptides (10 μL) at concentrations of 1–8 μM were mixed in a 96-well tissue culture plate. The mixture with bacteria and peptides was incubated for 24 h at 37 °C. After incubation, the supernatant was gently discarded, and the biofilms were fixed with 100% methanol for 10 min. The methanol was discarded and dried. The dried biofilms were stained with 0.1% crystal violet for 10 min and then rinsed with distilled water until the BHI medium supplemented with 0.2% glucose appeared colorless. Finally, the stained biofilms were dissolved in 95% ethanol and measured at an absorbance of 595 nm (A_595_). The percentage of biofilm mass was calculated using the following equation:Biofilm mass (%) = (A_595_ of treated AMPs/A_595_ of untreated biofilm) × 100

### 4.9. Circular Dichroism (CD) Spectroscopy

The CD spectra of Css54 at 40 μM were obtained in different buffers consisting of 10 mM sodium phosphate, 5 and 10 mM sodium dodecyl sulfate (SDS), and 20% and 40% 2,2,2-trifluoroethanol (TFE). CD spectra were measured using a Jasco 810 spectropolarimeter (Jasco, Tokyo, Japan) with a quartz cuvette (1.0-mm path length). The CD spectra were measured at wavelengths of 190–250 nm [52].

### 4.10. Outer Membrane Permeability Assay

Outer membrane permeability was measured using an NPN uptake assay. *Salmonella typhimurium* (ATCC 14028) was cultured in MHB medium. The bacteria were washed three times with 5 mM HEPES buffer (pH 7.2) and resuspended to an OD_600_ of 0.4 in 5 mM HEPES buffer (pH7.2). The bacteria were placed in a black 96-well plate, and NPN at 10 μM was added. Additionally, Css54 and melittin at 0.5×, 1×, and 2× MIC were added to each well. NPN fluorescence intensity was measured at excitation and emission wavelengths of 350 and 420 nm, respectively, every 5 min for 30 min.

### 4.11. Membrane Depolarization Assay

Membrane depolarization by the peptides in *L. monocytogenes* (KCTC 3710) and *S. typhimurium* (ATCC 14028) was measured using DiSC_3_(5). *Listeria monocytogenes* (KCTC 3710) and *Salmonella typhimurium* (ATCC 14028) were cultured in BHI and MHB media, respectively. The bacteria were washed three times with 5 mM HEPES buffer (pH 7.2) supplemented with 20 mM glucose. The bacteria were resuspended to an OD_600_ of 0.2 in 5 mM HEPES buffer (pH 7.2) supplemented with 20 mM glucose and 5 mM KCl. DiSC_3_(5) at 0.1 μM was added and placed in a black 96-well plate. The mixture was incubated for 1 h to stabilize the fluorescence intensity. Css54 and melittin at 0.5×, 1×, and 2× MIC were added to the mixture. The fluorescence intensity was measured at excitation and emission wavelengths of 622 and 670 nm, respectively [53].

### 4.12. SYTOX Green Uptake Assay

*Listeria monocytogenes* (KCTC 3710) and *Salmonella typhimurium* (ATCC 14028) were cultured in BHI and MHB media, respectively. The bacteria were washed three times with 10 mM Sp buffer (pH 7.2). The bacteria were resuspended to a final concentration of 2 × 10^7^ CFU/mL in 10 mM Sp buffer. The bacteria were incubated with 1 μM SYTOX Green for 15 min in a black 96-well plate. Css54 and melittin at 0.5×, 1×, and 2× MIC were added to the mixture. The fluorescence intensity was measured at excitation and emission wavelengths of 485 and 520 nm, respectively [54].

### 4.13. Flow Cytometry

The integrity of the bacterial membranes was analyzed by flow cytometry. *Listeria monocytogenes* (KCTC 3710) and *Salmonella typhimurium* (ATCC 14028) were cultured in BHI and MHB media, respectively. The bacteria were washed three times with 10 mM Sp buffer and then resuspended to an OD_600_ of 0.4 in 10 mM Sp buffer. The bacterial suspension was mixed with 2 μg/mL PI and incubated with Css54 at 0.5× and 1× MIC for 10 min. The suspension was centrifuged and washed to remove unbound dye. The data were measured by CytoFLEX flow cytometry (Beckman Coulter, Brea, CA, USA) [55].

## 5. Conclusions

In conclusion, Css54 has antimicrobial activity against bacteria in zoonotic disease. Css54 showed lower cytotoxicity than melittin. Moreover, Css54 maintained antimicrobial activity under the growth conditions of *L. monocytogenes* and showed thermal stability. Css54 effectively inhibited biofilm formation by *L. monocytogenes*. Css54 displayed an α-helix structure in bacterial membrane-mimicking environments. Css54 showed membrane lytic mechanisms similar to melittin. Our results suggest that Css54 can be used as an antibiotic and feed additive.

## Figures and Tables

**Figure 1 antibiotics-09-00831-f001:**
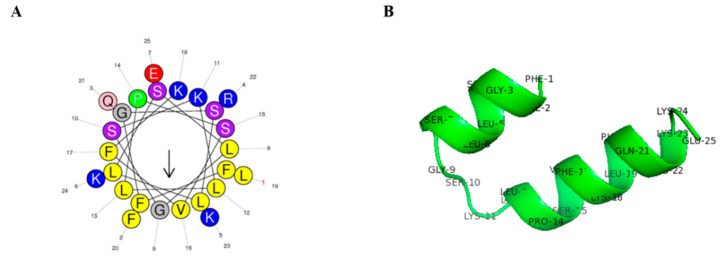
Structure analysis of Css54. (**A**) Helical wheel diagram of Css54 was obtained from the HeliQuest site (https://heliquest.ipmc.cnrs.fr/cgi-bin/ComputParams.py). (**B**) Three-dimensional structure of Css54 was predicted by PEP-FOLD3 and displayed by PyMOL.

**Figure 2 antibiotics-09-00831-f002:**
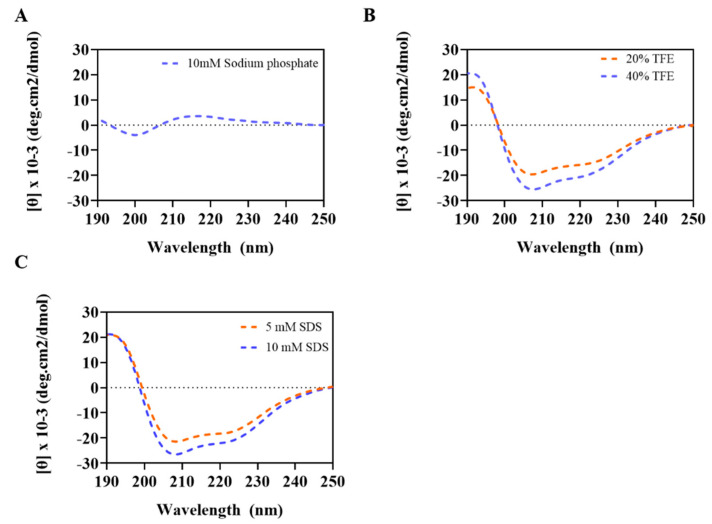
Circular dichroism (CD) spectra of Css54 measured in (**A**) 10 mM sodium phosphate, (**B**) TFE at 20% and 40%, and (**C**) SDS at 5 and 10 mM. Peptide concentration was fixed at 40 μM.

**Figure 3 antibiotics-09-00831-f003:**
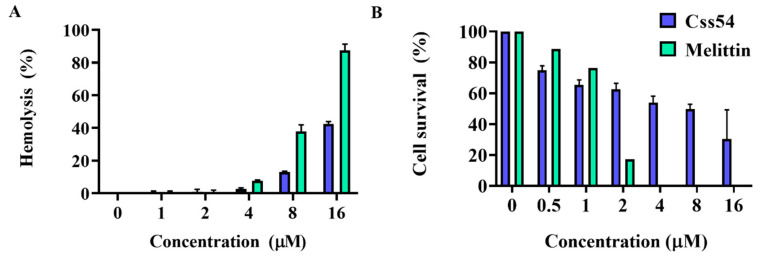
Hemolysis and cytotoxicity of antimicrobial peptides (AMPs). (**A**) Hemolytic activities were determined by measuring the release of hemoglobin from sRBCs at an absorbance wavelength of 414 nm. Percentage of hemolysis in sheep RBCs incubated with peptides at increased concentrations. (**B**) Cytotoxicity assay of AMPs. PK(15) cells isolated from porcine kidney were treated with Css54, and melittin at different concentrations in DMEM supplemented 10% FBS. Cell survival rates were measured by MTT assay.

**Figure 4 antibiotics-09-00831-f004:**
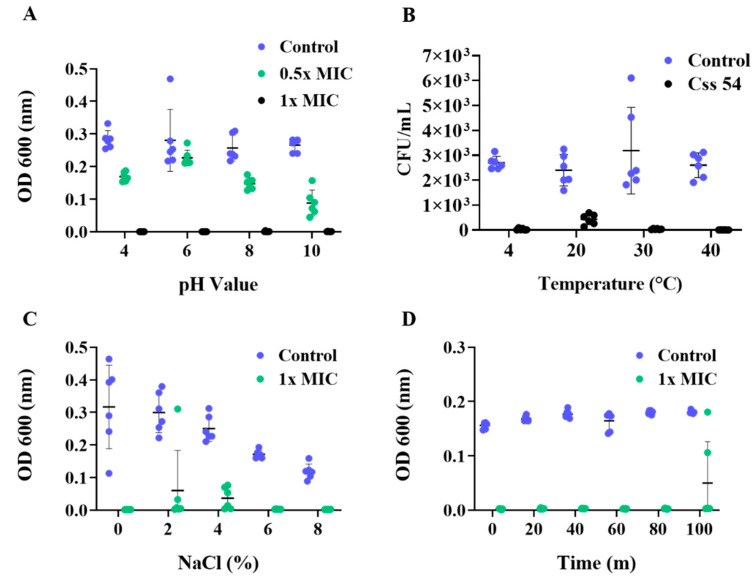
Antimicrobial activity of Css54 in different environments. (**A**) Effect of different pH values, (**B**) salt concentrations, and (**C**) temperatures on antimicrobial activity of Css54. (**D**) To evaluate thermal stability, Css54 at 1× MIC was incubated for 20, 40, 60, 80, and 100 min and then *L. monocytogenes* was added to the solution for incubation overnight at 37 °C.

**Figure 5 antibiotics-09-00831-f005:**
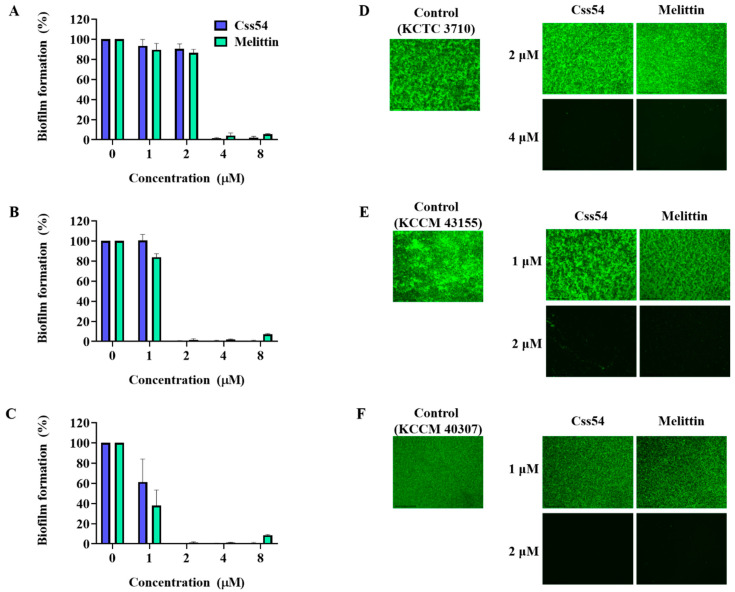
Inhibition of *L. monocytogenes* biofilm formation by Css54. (**A**) *L. monocytogenes* (KCTC 3710), (**B**) *L. monocytogenes* (KCCM 43155), and (**C**) *L. monocytogenes* (KCCM 40307) strains were mixed with AMPs for 24 h. The mass of each biofilm was determined by crystal violet staining. Fluorescence microscopy image of biofilms formed by (**D**) *L. monocytogenes* (KCTC 3710), (**E**) *L. monocytogenes* (KCCM 43155), and (**F**) *L. monocytogenes* (KCCM 40307) strains after treatment with AMPs. Live bacteria were stained with SYTO9 dye and analyzed with an EVOS FL Auto 2 imaging system (Invitrogen).

**Figure 6 antibiotics-09-00831-f006:**
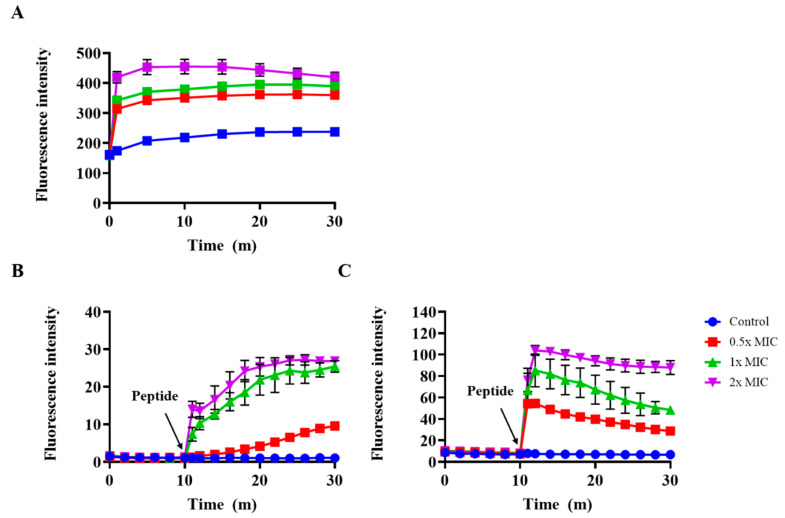
Outer membrane permeability and membrane depolarization assay. (**A**) *S. typhimurium* (ATCC 14028) outer membrane permeability induced by peptides. The outer membrane permeability of *S. typhimurium* induced by Css54 was measured using the fluorescent dye NPN (*N*-phenyl-1-naphthylamine. (**B**) *S. typhimurium* and (**C**) *L. monocytogenes* cytoplasmic membrane depolarization assay. Bacterial cytoplasmic membrane depolarization was analyzed using the potential-sensitive dye DiSC_3_(5). Various concentrations of Css54 were added to the bacteria.

**Figure 7 antibiotics-09-00831-f007:**
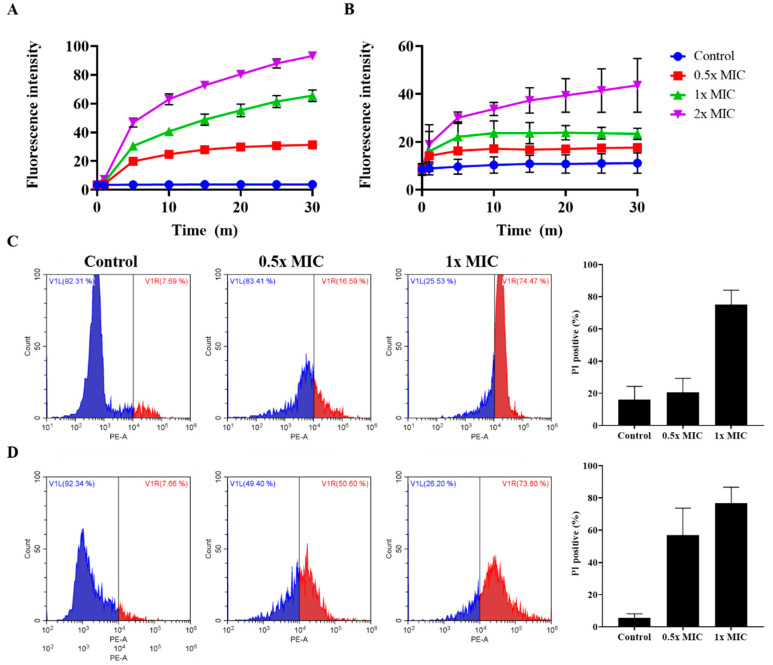
Analysis of *L. monocytogenes* and *S. typhimurium* membrane integrity after treatment with Css54. Css54 at 0.5×, 1×, and 2× MIC were added to (**A**) *S. typhimurium* and (**B**) *L. monocytogenes*. Fluorescence intensity of SYTOX green was measured at emission and excitation wavelengths of 520 and 485 nm, respectively. Flow cytometry analysis using propidium iodide staining against bacterial membrane integrity. Changes in (**C**) *S. typhimurium* and (**D**) *L. monocytogenes* after treatment with peptide at 0.5× and 1× MIC for 10 min were measured by flow cytometry using propidium iodide (PI).

**Table 1 antibiotics-09-00831-t001:** Amino acid sequence and properties of Css54.

Name	Sequence	Molecular Mass (Da)	Net Charge	H ^a^
Css54	FFGSLLSLGSKLLPSVFKLFQRKKE-NH_2_	2869.5	5	0.451

^a^ H means hydrophobicity.

**Table 2 antibiotics-09-00831-t002:** Minimum inhibitory concentrations (MICs) of antimicrobial peptides and antibiotics against zoonosis bacteria.

Strains	MICs (μM)					
	Css 54	Melittin	Cefotaxime	Vancomycin	Colistin	Ampicillin
Gram-positive (+)						
*Listeria monocytogenes*	2	2	64	0.5	64	1
(KCTC 3710)						
*Listeria monocytogenes*	2	1	4	0.5	32	0.5
(KCCM 40307)						
*Listeria monocytogenes*	2	1	4	0.5	16	1
(KCCM 43155)						
*Streptococcus suis*	2	2	<0.13	<0.13	>64	0.5
(KCTC 3557)						
Gram-negative (−)						
*Campylobacter jejuni*	2	2	32	0.5	>64	>64
(KCTC 5327)						
*Salmonella typhimurium*	4	2	1	64	1	2
(ATCC 14028)						
*Salmonella typhimurium*	4	2,	0.5	64	2	>64
(CCARM 8009)						
*Salmonella typhimurium*	4	2	0.5	64	4	>64
(CCARM 8013)						

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
