# Peer review of "Scorpion-Venom-Derived Antimicrobial Peptide Css54 Exerts Potent Antimicrobial Activity by Disrupting Bacterial Membrane of Zoonotic Bacteria"

_antibiotics, 2020, doi:10.3390/antibiotics9110831_

Round 1

Reviewer 1 Report

The manuscript entitled "Scorpion Venom-derived Antimicrobial Peptide Css54 Exerts Potent Antimicrobial Activity by Disrupting Bacterial Membrane of Zoonotic Bacteria" describes the antimicrobial activity and mechanism of action of the peptide toxin Css54 against a variety of bacteria causing zoonotic diseases. The work is concise, very well written, within the journal's scope, and should be interesting for the general reader.
However, I would like to point out several issues that, in my opinion, need to be addressed:
1-Since a different research group originally isolated Css54, the authors of the present work should present their data (supplementary materials) such as chromatogram(s), molecular mass measurement, and sequence analysis to demonstrate the presence of Css54. These data are critical as this work entirely relies on a known molecule that must be pure and unmistakenly identified.
2- Animal venoms contain a large number of harmful peptide neurotoxins and hemolysins. Css54 is a toxin that comes from spider venom; it is very resistant to heat and might be resistant to degradation. In lines 430-431, it is written: "Our results suggest that Css54 can be used as an antibiotic and feed additive". I think this statement may need more experimental evidence or rewriting (softened) since a feed additive should not be toxic if ingested.
Minor comments:
-Please briefly describe in Figure 1 how the structural model obtained by PEP-FOLD3 was assessed. Also include references for PEP-FOLD3 and Pymol.
-Line 80: change "veneom" for venom.

Author Response

Response to Reviewer 1:

Comments and Suggestions for Authors:
The manuscript entitled "Scorpion Venom-derived Antimicrobial Peptide Css54 Exerts Potent Antimicrobial Activity by Disrupting Bacterial Membrane of Zoonotic Bacteria" describes the antimicrobial activity and mechanism of action of the peptide toxin Css54 against a variety of bacteria causing zoonotic diseases. The work is concise, very well written, within the journal's scope, and should be interesting for the general reader.
However, I would like to point out several issues that, in my opinion, need to be addressed:

  1. Since a different research group originally isolated Css54, the authors of the present work should present their data (supplementary materials) such as chromatogram(s), molecular mass measurement, and sequence analysis to demonstrate the presence of Css54. These data are critical as this work entirely relies on a known molecule that must be pure and unmistakenly identified.

Reply: Thank you for this comment. As the reviewer suggested, we have added the RP-HPLC profile and MALDI mass spectrometric analysis to the supplementary data.

(After revision)

Lines 85–87

Css54 synthesis and molecular weights were confirmed by reversed-phase high-performance liquid chromatography (RP-HPLC) using a C18 column and matrix-assisted laser desorption ionization time-of-flight mass spectrometry (MALDI-TOF MS) (Figure S1A and B).

Supplementary data

  • Figure S1. RP-HPLC and mass spectrometry. (A) RP-HPLC profile on a C18 column with detection at 230 nm. The black arrow represents the retention time of Css54 (25 min). (B) MALDI mass spectrometric analysis of Css54. The respective mass/charge ratio was 2869.5.

  1. Animal venoms contain a large number of harmful peptide neurotoxins and hemolysins. Css54 is a toxin that comes from spider venom; it is very resistant to heat and might be resistant to degradation.

Reply: We agree with the reviewer’s comment. The heat stability of antimicrobial peptides is an important characteristic affecting their application potential. Various procedures, such as feed or food processing, include a heating step. Therefore, we confirmed that Css54 retained its antimicrobial activity for 80 min but this activity was lost by 100 min after heating at 100°C (Figure 4D).

(Reply reference)

  1. Dong, N.; Chou, S.; Li, J.; Xue, C.; Li, X.; Cheng, B.; Shan, A.; Xu, L. Short symmetric-end antimicrobial peptides centered on β-turn amino acid units improve selectivity and stability. Front Microbiol 2018, 9, 2832.
  2. Ma, Q.-Q.; Dong, N.; Shan, A.-S.; Lv, Y.-F.; Li, Y.-Z.; Chen, Z.-H.; Cheng, B.-J.; Li, Z.-Y. Biochemical

property and membrane-peptide interactions of de novo antimicrobial peptides designed by helix-

forming units. Amino Acids 2012, 43, 2527-2536.

  1. In lines 430-431, it is written: "Our results suggest that Css54 can be used as an antibiotic and feed additive". I think this statement may need more experimental evidence or rewriting (softened) since a feed additive should not be toxic if ingested.

Reply: We agree with the reviewer’s comment. Melittin, a component of bee venom, has antimicrobial activity but shows toxicity in mammalian cells. However, melittin has many biological effects such as antimicrobial activity and antitumor effects. We confirmed that Css54 has lower toxicity than melittin but has similar antibacterial activity. To address the reviewer’s comment, we have cited several previously reported publications and added information to the Discussion.

(After revision)

Lines 252–257

In a previous study, treatment with melittin showed no adverse effects on the stomach tissue, including on its function, and melittin was confirmed to have a protective effect against gastric inflammation and antitumor effects in vivo [39-41]. These reports indicate that Css54 can be developed as an effective treatment because of its lower toxicity than melittin. We confirmed that hemolytic activity towards sRBCs and cytotoxicity against pig kidney cells PK(15) was lower than that of melittin.

References

  1. Abu-Zinadah, O.; Rahmy, T.; Alahmari, A.; Abdu, F. Effect of melittin on mice stomach. Saudi J Biol Sci 2014, 21, 99-108.
  2. Rahmy, T.; Alahmari, A.; Abdu, F.; Abu-Zinadah, O. Protective effect of melittin against gastric inflammation in mice. Life Sci J 2013, 10.
  3. Duffy, C.; Sorolla, A.; Wang, E.; Golden, E.; Woodward, E.; Davern, K.; Ho, D.; Johnstone, E.; Pfleger, K.; Redfern, A. Honeybee venom and melittin suppress growth factor receptor activation in her2-enriched and triple-negative breast cancer. NPJ Precis Oncol 2020, 4, 1-16.

Minor comments:
1. 1. Please briefly describe in Figure 1 how the structural model obtained by PEP-FOLD3 was assessed. Also include references for PEP-FOLD3 and Pymol.

Reply: Thank you for this comment. As the reviewer suggested, we have added an explanation and references.

(After revision)

Lines 328–331

Projections of the predicted three-dimensional structures were confirmed using PEP-FOLD3 (https://bioserv.rpbs.univ-paris-diderot.fr/services/PEP-FOLD3/). PEP-FOLD3 can be used for structural characterization of the peptides, followed by visualization using PyMOL [48, 49].

References

  1. Lamiable, A.; Thévenet, P.; Rey, J.; Vavrusa, M.; Derreumaux, P.; Tufféry, P. Pep-fold3: Faster de novo structure prediction for linear peptides in solution and in complex. Nucleic Acids Res 2016, 44, W449-W454.
  2. Yuan, S.; Chan, H.S.; Hu, Z. Using pymol as a platform for computational drug design. Wiley Interdiscip Rev Comput Mol Sci 2017, 7, e1298.

2.Line 80: change "veneom" for venom.

Reply: Thank you for pointing this out. This error has been corrected.

(After revision)

Line 81

Css54 was isolated from C. suffusus venom.

Reviewer 2 Report

Generally I find the presented research decently described and the manuscript well written. I have no major remarks.

Regarding bacterial species, please make clear in the introduction for which species Css54 has been tested so far and with what outcome.

The statement "Css54 exhibited low cytotoxicity compared to melittin", appearing repeatedly in the manuscript, is a bit misleading as suggesting "low" Css54 toxicity. I surmise "lower" is what the authors had in mind. Anyway, for the concentrations used, PK(15) cells exhibit c. 60% survival. Doesn't it make Css54 quite cytotoxic after all? In my opinion it requires detailed discussion in the manuscript.

Figure 6B and C: why is there 10 min lag before any change is observed? Could you clarify it in the figure description and manuscript text?

Author Response

Response to Reviewer 2:

Generally, I find the presented research decently described and the manuscript is well written. I have no major remarks.

  1. Regarding bacterial species, please make clear in the introduction for which species Css54 has been tested so far and with what outcome.

Reply: We agree with the reviewer’s comment. We have already described the previously reported antimicrobial activity of Css54 against Staphylococcus aureus and Escherichia coli (lines 68–69). Moreover, we described which strain was used to test the antimicrobial activity of Css54 in the Introduction section according to the reviewer’s comment.

(After revision)

(Lines 70–73)

In this study, Css54 was tested for its antimicrobial activity against L. monocytogenes, S. typhimurium, Streptococcus suis, and Campylobacter jejuni, which cause zoonotic disease. We used melittin, an antimicrobial peptide found in bee venom. We confirmed that Css54 has antimicrobial activity  similar to melittin.

  1. The statement "Css54 exhibited low cytotoxicity compared to melittin", appearing repeatedly in the manuscript, is a bit misleading as suggesting "low" Css54 toxicity. I surmise "lower" is what the authors had in mind.

Reply: Thank you for this comment. We have revised these sentences.

(After revision)

(Line 17)

the cytotoxicity and hemolytic activity of Css54 was lower than that of melittin isolated from bee venom.

(Line 127)

Css54 exhibited lower cytotoxicity than melittin.

(Line 438)

Css54 showed lower cytotoxicity than melittin.

  1. Anyway, for the concentrations used, PK(15) cells exhibit c. 60% survival. Doesn't it make Css54 quite cytotoxic after all? In my opinion it requires detailed discussion in the manuscript.

Reply: Thank you for your comment. As the reviewer suggested, we have further explained this point in the Discussion section.

(After revision)

(Lines 254–259)

In a previous study, treatment with melittin showed no adverse effects on the stomach tissue, including on its function, and melittin was confirmed to have a protective effect against gastric inflammation and antitumor effects in vivo [39-41]. These reports indicate that Css54 can be developed as an effective treatment because of its lower toxicity than melittin. We confirmed that hemolytic activity towards sRBCs and cytotoxicity against pig kidney cells PK(15) was lower than that of melittin.

References

  1. Abu-Zinadah, O.; Rahmy, T.; Alahmari, A.; Abdu, F. Effect of melittin on mice stomach. Saudi J Biol Sci 2014, 21, 99-108.
  2. Rahmy, T.; Alahmari, A.; Abdu, F.; Abu-Zinadah, O. Protective effect of melittin against gastric inflammation in mice. Life Sci J 2013, 10.
  3. Duffy, C.; Sorolla, A.; Wang, E.; Golden, E.; Woodward, E.; Davern, K.; Ho, D.; Johnstone, E.; Pfleger, K.; Redfern, A. Honeybee venom and melittin suppress growth factor receptor activation in her2-enriched and triple-negative breast cancer. NPJ Prec Oncol 2020, 4, 1-16.

  1. Figure 6B and C: Why is there a 10 min lag before any change is observed? Could you clarify this in the figure description and manuscript text?

Reply: Thank you for this question. We confirmed that bacterial membrane depolarization occurred after treatment with Css54 using 3,3’-dipropylthiadicarbocyanine iodide DiSC3(5). DiSC3(5) accumulates in the lipid bilayer, causing self-quenching, and is released upon damage to the cytoplasmic membrane, resulting in increased fluorescence intensity. As mentioned in the Methods section, bacteria treated with DiSC3(5) were incubated for 1 h to stabilize the fluorescence intensity. The reason for adding Css54 after 10 min was to ensure that the fluorescence intensity had been stabilized. To address the reviewer’s comment, we have revised Figure 6B and C.

References

  1. Sun, Y.; Dong, W.; Sun, L.; Ma, L.; Shang, D. Insights into the membrane interaction mechanism and antibacterial properties of chensinin-1b. Biomaterials 2015, 37, 299-311.

2              Kim, M.K.; Kang, N.H.; Ko, S.J.; Park, J.; Park, E.; Shin, D.W.; Kim, S.H.; Lee, S.A.; Lee, J.I.; Lee, S.H. Antibacterial and antibiofilm activity and mode of action of magainin 2 against drug-resistant acinetobacter baumannii. Int J Mol Sci 2018, 19, 3041.

(After revision)

(Line 194)

 [E1]Do you mean that this peptide has antimicrobial activity?

Round 2

Reviewer 1 Report

The authors have successfully addressed the question raised previously.
I would like to add a comment for a better understanding of the Results sections. Line 82: "Css54 was isolated from C. suffusus venom." After reading the new version, I understood from the new manuscript that the authors only synthesized the peptide and did not isolate it from the venom. In any case, if the authors only used the synthetic peptide, I suggest removing that sentence from Line 82 since it is confusing. I think it is not necessary to clarify the natural origin of Css54, since this was already mentioned in the Introduction (reference number 20).